# Neighborhood Social and Built Environment and Disparities in the Risk of Hypertension: A Cross-Sectional Study

**DOI:** 10.3390/ijerph17207696

**Published:** 2020-10-21

**Authors:** Regina Grazuleviciene, Sandra Andrusaityte, Tomas Gražulevičius, Audrius Dėdelė

**Affiliations:** Department of Environmental Science, Vytautas Magnus University, K. Donelaicio 58, 44248 Kaunas, Lithuania; sandra.andrusaityte@vdu.lt (S.A.); t.grazulevicius@gmail.com (T.G.); audrius.dedele@vdu.lt (A.D.)

**Keywords:** socioeconomic status, neighborhoods, disparities, hypertension, citizen science, epidemiological study

## Abstract

Citizens’ participation in urban environmental quality assessment is important when identifying local problems in the sustainable development and environmental planning policy. The principal aim of this study was to analyze whether any social differences exist between the joint effect of built neighborhood quality and exposure to urban green spaces and the risk of hypertension. The study sample consisted of 580 participants residing in 11 districts in Kaunas city, Lithuania. Using geographic information systems (GIS), individual data on the socioeconomic status (SES) and health were linked to the participants’ perceptions of the environmental quality and exposure to green spaces (NDVI). We used multivariate logistic regression to estimate associations as odds ratios (OR). Those study participants with lower education and those study participants with higher education on low incomes rated their health significantly worse. Low SES persons residing in areas with low exposure to green spaces had a significantly higher risk of hypertension when sex, age, family status, smoking, and income were accounted for (OR 1.83, 95% CI 1.01–3.36). This citizen science study provided evidence that the social environment and the quality of the built environment had a complex effect on disparities in the risk of hypertension.

## 1. Introduction

Over the past years, as the news of the UN program of sustainable development has spread, people’s interest in the quality of the living environment and health issues has increased significantly. Public engagement in the research activities through citizen science could make important contributions to societal transformations and could exert pressure on politicians to solve environmental problems and health disparities [1]. Environmental health problems are increasingly recognized as a major contributor to chronic diseases and premature deaths. It was indicated that both scientists and public participants of citizen science would benefit from public engagement by increasing communities’ awareness and input in identifying local problems, the usage of new technology, and education on issues of the environment and health. Scientists and public partnership would have the potential to influence policy devoted to social environment and citizens’ health [2].

Social determinants of health played an important role in addressing the sustainable development goals targets to promote health and well-being and to reduce disparity [3]. The social environment, comprising personal and societal relationships, institutions, cultures, and physical surroundings—all the aspects and products of human activity and interaction—is insufficiently researched in relation to health.

There is some evidence that the socioeconomic status (SES) defined by education, occupation, or income, underlies major determinants of health, such as environmental exposure, health behavior, chronic stress, and health care. SES is associated with cardiovascular disease, hypertension, diabetes, and other chronic non-communicable disease [4,5]. However, it is still very unclear which segment of the SES actually has the biggest impact on health—financial aspects such as income, or non-financial ones such as education, or the social environment. It is also not clear what effect the neighborhood social and built environment has on the disparities in the prevalence of hypertension.

Identification of these mechanisms would provide evidence for the necessary actions and policy measures for the reduction of modifiable environmental risks and the elimination of health disparities.

Some study findings have provided evidence that neighborhood environment has an independent influence on health outcomes, such as cardiovascular disease and hypertension—the second leading cause of chronic diseases worldwide [6,7]. Hypertension has been identified as an environment-related disease, and social environment factors may be modifiable through environmental policy changes or health behavior changes [8,9,10]. The impact of SES on hypertension has been reported in several studies with conflicting results [11]. In addition, environmental epidemiological studies of the relationships between neighborhood environment, its perception and hypertension are sparse, and the associations with different indicators of the social environment effects remain uncertain [12,13,14]. Therefore, evidence of associations between the risk of hypertension and the social environment disparities may provide new data for the prevention of chronic diseases and the implementation of sustainable development goals.

The majority of the environmental studies that explored the relationship between greenness and blood pressure used a cross-sectional design, but did not include citizens’ perception of the neighborhood quality. Therefore, the evidence concerning the effects of greenness on hypertension remains mixed and inconclusive [12,15]. These studies used the community area classification method as an index of green space exposure, and very few of these studies examined the complex effects of the social environment and the built environment [16,17], or analyzed the influence of neighborhood characteristics on residents’ perceptions not including the health effects [18,19]. 

This study has been initiated considering citizens’ needs and has been aimed at evaluating the impact of the complex effects of the built and social environment on health. We investigated the citizens’ perception of the residential district or neighborhood quality and the social influences of their personal exposure to green spaces on hypertension. The citizens’ perception of infrastructure and our study results can influence the LAND use pattern and may provide information for sustainable cities and communities regarding urban planning and urban management problems [20,21,22].

The present study is part of the Horizon 2020 proposal Citizen Science for Urban Environment and Health (CitieS-Health) [22]. Our previously publicized findings showed that the poor quality of the neighborhood and the individual-level characteristics were those very factors that influenced a higher prevalence of health problems at the district level [23]. This study had three aims: (1) to estimate whether the citizens’ concerns, neighborhood quality rating and the greenspace usage depended on the education level; (2) to assess the relationships between joint individual SES markers (education and income) and the risk of hypertension; and (3) to test the hypothesis that the differences in the effect of the social environment on hypertension were mediated by exposure to urban green spaces.

## 2. Materials and Methods 

### 2.1. Study Design

This citizens’ collaborative study was conducted 2019–2020. The study sample consisted of 580 18–75-year-old participants residing in 11 districts of Kaunas city, Lithuania. The city covers 15,700 hectares, 8329 hectares of which are covered by greenery (parks, groves, gardens, natural reserves, and agricultural areas). Our definition of “city parks” includes areas larger than 1 ha, with 65% of land covered with trees. All Kaunas city parks are open to the public, are located amidst residential homes or establishments and near to the public transport lines, and offer some recreation opportunities (e.g., walking, jogging, rollerblading, physical training, or resting on the bench).

Our study participants’ engagement methods included radio information, announcements in local newspapers and web sites, advertisements at community events, and conferences. Since the citizens’ response to the invitation via mass media was poor, we also used the list of the participants of the scientific-practical conference Human and Nature Safety to send personal invitations to join the study. From the beginning of the research project, 10 participants were involved in the discussions by identifying and anticipating environmental and health issues, planning the study design, and discussing the study questionnaire.

On 9 May 2019, we organized an open meeting with the general population, the study participants, and scientists. The aim of the meeting was to obtain new information on citizens’ concern about environmental and health issues. In total, 120 participants took part in the meeting, among them researchers, citizens, journalists, NGO representatives, and a planning and architecture specialist of Kaunas city. The participants signed the informed consent form and filled out the formalized questionnaire. The questionnaire had closed-ended and open-ended questions. On 20 May 2019, we organized a meeting to formulate research questions and to create the study protocol. The participants of the meeting were scientists, citizens’ representatives, and a planning and architecture specialist of Kaunas city (in total, 21 participants). During the debates, we discussed citizens’ environmental concerns and major health concerns, formulated the research questions as well as the aim of the study and the study protocol. The collaborative project benefited from major environmental concerns that the citizens named, such as air pollution, traffic noise, and the availability of cycling and smooth walking paths. The major health concerns mentioned by the participants were hypertension, obesity, and cardiac problems. A detailed description of the methods of the participants’ involvement, as well as the description of the study, have been provided previously [23]. All participants gave their informed consent for inclusion before entering the study. The study was conducted in accordance with the Declaration of Helsinki [24]. The study protocol, the questionnaire, and the consent procedure were approved by the Kaunas Regional Committee for Biomedical Research Ethics (BE-2-51. 2019-06-10). Using environmental epidemiological approach, we have sought to answer the following study participants’ question: “Why do citizens in my district suffer from hypertension more often than those in other ones?” To answer this question, we conducted a cross-sectional study. We conducted an analysis of our survey participants linked to GIS data describing their neighborhoods. We used geospatial analysis and adjusted multi-level models to assess whether any social differences existed in the effect of the objectively measured and subjectively measured (using questionnaire) quality of the built neighborhood and exposure to urban green spaces on the prevalence of hypertension by controlling the influence of possible confounding variables.

### 2.2. Measurements

In this study, we objectively measured environmental exposure to green space by estimation of normalized difference vegetation index (NDVI) for every participant’s home address. Traffic flows and green space NDVI were linked to each individual participant’s place of residence. We also collected data on baseline characteristics and subjective rating on residential district exposures by using standardized questionnaires. To obtain the participants’ perception on environmental quality, the participants rated questions about their neighborhood, built environment, and social environment. The answers were scored using a seven-point Likert rating scale, which included a series of questions to be answered in points ranging from 1 (strongly disagree) to 7 (strongly agree) in order to measure mean environmental perceptions. Seven-point Likert items have been shown to be more accurate, easier to use, and better reflect a respondent’s true evaluation [25]. A 1–7 scale gives two levels between “disagree strongly” and neutral. Higher scores indicated better neighborhood conditions. Information collected through questionnaires and interviews also covered socio-demographic and health-related data as follows: age, sex, self-reported health, behavior, SES, and residence history.

The participants’ perception of the characteristics of the social environment was assessed using questions on the infrastructure in the residence neighborhood: public transport, pathways and cycling routes, walking distances to the city’s green spaces or parks, and areas adapted for exercise and relaxation. There were questions on the opportunities for societal relationships, social well-being: public spaces to meet people, stress level, social cohesion and inclusion among different groups and the feeling of safety in the place of residence. In addition, we asked questions on environmental quality such as problems caused by air pollution and noise in the place of residence, and on the average time spent outdoors per day for fast walking, biking, or gardening.

We assessed individual-level predictors of the SES by evaluating the participants’ education level, situation at work, and income. The education level was ascertained in years and classified in levels as secondary (less than high school or high school), college, and university. In the analysis, we dichotomized the education level and used lower education and higher education (university) groups. The occupational status was ranked as low (a blue-collar worker), medium (a housekeeper or an officer), and high (a manager or a company owner), and the monthly net income was also ranked as low (less than 400 Euros) or higher (400 Euros or more).

Hypertension was defined using the European Society of Cardiology and European Society of Hypertension criteria [26] by the presence or absence of physician-diagnosed hypertension, the reported use of antihypertensive medication, and/or systolic blood pressure of 140 mmHg or higher and/or diastolic blood pressure of 90 mmHg or higher. The accuracy and consistency of the study participants’ reporting of physician-diagnosed hypertension was based on a comparison of data with responses on blood pressure readings, and the consistency of answers on time spent outdoors was based on comparisons with time spent in a park.

Body mass index (BMI) was defined as weight (kilograms) divided by squared height (meters) (kg/m^2^). The participants with the BMI above 30 were classified as obese. Smoking status was self-reported as never, past, or current. We collected information on physical activity during the last week obtaining participants’ feedback about the mean time per day spent outdoors for fast walking, biking, or gardening. The participants were then classified into two groups according to the Public Health Guidelines for Physical Activity [27], i.e., at least 150 min/week of moderate-intensity physical activity outdoors (the recommended duration), or fewer min/week spent outdoors. The participants residing in the street with above 10,000 cars/day were classified as exposed to heavy traffic emissions in their place of residence.

In the statistical analysis, we dichotomized personal data and used mean values of environmental perception scores as cut points for an easier interpretation of logistic regression estimates. The assessment of residential greenness was based on a normalized difference vegetation index (NDVI) that was derived from Landsat 7 Enhanced Thematic Mapper Plus (ETM+) data at 30 × 30 m resolution [28].

We measured NDVI for every participant’s home address at a buffer distance of 100 m, and the estimate was used to calculate the residential district mean NDVI using estimates of all the geocoded district residents’ addresses. The residential addresses at the time of the interview were geocoded according to the street name and the house number, and the maps of NDVI were generated using the image that was obtained on 10 July 2011. Mean NDVI values of all the study participants’ home addresses were calculated as estimations of the level of greenness within every residential district. The district green spaces measured with the NDVI included parks, groves, green roofs, streams, and community gardens.

### 2.3. Statistical Analysis

First, frequency distributions of participant characteristics were tabulated for the analytic sample. We used the chi-squared test to compare the values and the frequencies of the participants’ baseline characteristics and perceptions of neighborhood quality by education level. Quantitative variables were reported as mean values and standard deviations. Statistical significance was set at *p*-value < 0.05. Second, we applied multivariate logistic regression analysis to evaluate the associations between individual SES and the risk of hypertension as odds ratios (OR) and their 95% confidence intervals (CI). In order to estimate the relationships between education level, greenness exposure, and the risk of hypertension, we used multivariate logistic regression models and a stratified analysis controlling the variables that significantly influenced the relationship between SES and hypertension.

In the multivariate logistic regression models, we applied higher than 0.05 *p*-value thresholds (such as <0.2) for the inclusion of predictor variables from bivariate statistics in order to prevent the exclusion of relevant factors [29,30]. For this reason, we also retained the variables that changed the adjusted odds ratios (aOR) by 10% or more for inclusion in the multivariate logistic regression analysis. We used Fisher’s exact tests to compare the qualitative characteristics between the groups. A multivariate logistic model was used in order to determine the independent association between hypertension and environmental exposure controlling for possible confounding variables such as sex, education level, age, smoking status, and income.

In addition, we conducted geospatial analysis using the ArcGIS 10.4 mapping and analytics platform (Esri, Redlands, CA, USA) to assess the associations among the unadjusted prevalence of hypertension, the presence or absence of a major road in the residence area, and the mean exposure to green spaces (NDVI) among participants residing in different districts. Significance was accepted at an alpha level of 0.05. Statistical analyses were performed using SPSS version 25.0 package (IBM Corporation, New York, NY, USA).

## 3. Results

### 3.1. Participants’ Characteristics by Education Level

In order to identify personal characteristics associated with education level, we performed an unadjusted factor analysis by lower (secondary and college) and higher (university) levels of education (Table 1 and Table 2).

There were no significant differences in education level among the three age groups of the participants, among participants of different residence districts and exposure to traffic flows, or among participants with different duration of living in the current place of residence. However, significant differences in education level were found among the participants regarding sex, marital status, situation at work, and monthly net income. Participants with a higher education level were more often were women, married, and full-time employees with higher monthly net income.

Table 2 shows the unadjusted prevalence of self-reported health characteristics and physical activity behavior in groups of the participants with lower and higher levels of education. In Kaunas city, the mean prevalence of hypertension was 33.3%. The prevalence of hypertension did not differ significantly between the two education level groups (*p* = 0.157). However, the participants in the lower education group had a higher mean systolic blood pressure (*p* = 0.024) and diastolic blood pressure (*p* = 0.002), and were obese more often (*p* = 0.029), compared to those of the higher education group. Chronic diseases were reported by 32.2% of the participants in the lower education group compared to 22.7% of better-educated participants (*p* = 0.012). The prevalence of poor health status and smoking was significantly higher among the participants of the lower education group. The self-reported physical activity outdoors was low in both groups, mostly not reaching the level recommended by the public health guidelines for physical activity, i.e., at least 150 min/week of moderate-intensity physical activity. Only 10.7% of the participants in the lower education group and 12.9% of those in the higher education group (*p* = 0.111) reached that level. Better educated participants more often visited the natural environment, but the difference was not statistically significant.

Our results showed that about 24.5% of the participants whose monthly net income was less than 400 euros had a higher education level. In order to reveal the specific characteristics of this higher education level whose monthly net income was less than 400 euros group of participants, we compared their self-reported health characteristics with those of the group of the participants with a higher education level whose income was higher than 400 euros. We found that the participants with higher education and low income had significantly higher systolic blood pressure, suffered from chronic disease and hypertension more often, and also rated their health as poor more often. These findings revealed that both low income and lower education levels were associated with a higher prevalence of health problems. The self-reported health characteristics in the group of the participants with hypertension significantly differed from those in the group of the participants without hypertension: the participants of the former group had a higher body mass index, worse situation at work, lower monthly net income, were smoking, had a higher prevalence of chronic diseases, and presented poorer health rating. The participants of the hypertension group more often resided on streets with over 10,000 cars/day for longer than 20 years (*p* = 0.033). There is a clear individual-level SES gradient in personal characteristics and health status.

### 3.2. Participants’ Ratings of the Neighborhood Quality and Social Well-Being by Education Level

We performed a univariate analysis to study the distribution of the ratings perceptions of environmental characteristics of the living district by education level. We used the mean environmental perceptions score to examine whether the participants’ perceptions of neighborhood quality and social well-being (assessed by the feeling of safety, social cohesion and inclusion and stress level, depended on their education level.

Higher scores indicated better ratings of the social neighborhood conditions. Table 3 shows the neighborhood quality ratings among the lower educated and higher educated participants. The participants of both (lower education and higher education) groups similarly (*p* > 0.05) highly rated the public transport in the district of residence, indicating that public transport met their needs (5.42 and 5.19, accordingly). The participants of both groups also acknowledged that there were good opportunities for walking to reach the city’s green spaces or parks (5.06 and 5.15, respectively), and pointed out that they felt safe in their residence area (5.03 and 5.26, respectively). There were no significant differences in stress, tension, or anxiety felt or in the satisfaction with pathways, cycling routes, and areas adapted for exercising and relaxation, as well as with the public spaces available for social interaction. Higher educated participants regularly visited the natural environment, more often than lower educated ones (4.41 and 4.00, respectively; *p* = 0.031). The question “Can you take part in decision-making to improve the environment in which you live?” was rated as the lowest in both groups of the participants, namely 3.50 in the group of participants with a higher education level and 3.14 in the group of participants with a lower education level, at *p* = 0.063, showing a potential for improving the social cohesion and inclusion situation between different groups.

In addition, the perception of environmental quality, such as problems caused by air pollution and noise in the place of residence, did not differ significantly between the two participant groups. The mean ratings of the perceptions of the neighborhood quality and social well-being did not depend on the education level. Our study results show that the indicator of well-being is the outcome of multiple environmental factors not limited to education level.

In urbanized areas, environmental quality may depend on a city’s green spaces that are important for physical activity, stress reduction, and health. We linked the green space estimates to every participant’s health data, seeking to identify whether higher green space levels (NDVI values) were associated with better health outcomes and the prevalence of hypertension. We estimated the mean NDVI of every residence district and the mean NDVI on the city level. The mean NDVI in Kaunas city was 0.571. Such greenness level shows an area containing a dense vegetation canopy. The districts with NDVI below the mean we treated as low greenness level districts, while districts above the mean we treated as high greenness level districts.

In order to map the participants’ concerns of the different prevalence of hypertension, we explored spatial patterning in NDVI (presented by the <mean or >mean) and the prevalence of hypertension (in percentage) at the district level. The prevalence of hypertension was estimated according to the self-reported data of the study participants, and a map of Kaunas city was created (Figure 1). The transport network of Kaunas city has an annular-radial structure. The unadjusted prevalence of hypertension in various districts ranged from 15.0% to 51.3%. However, these numbers do not represent the real situation of all the population of all Kaunas districts. A higher than mean (33.3%) prevalence of hypertension was found in five of 11 districts, mainly where traffic flows were high, and the participants were more concerned about the health consequences of poor air quality. In five of low greenness level districts (NDVI < mean), predominantly in the city center, the prevalence of hypertension was higher than the mean. However, in the districts with low exposure to green spaces located on the hills (Žaliakalnis and Dainava), Kaunas citizens were less concerned about the health consequences of poor air quality, and the prevalence of hypertension was lower. Meanwhile, the participants in the periphery of the city resided in greener areas, and the prevalence of hypertension was lower as well. It is important to bear in mind that disparities in the prevalence of hypertension across districts may be due to study participants’ characteristics rather than the characteristics of all districts’ populations because the study did not include samples of representatives of all Kaunas districts. These data were not adjusted for possible confounding variables such as personal age, sex, education level, smoking, or income. Therefore, we were not able to explain the reasons of unequal prevalence of hypertension in Kaunas districts.

### 3.3. The Relationships Between Individual SES and the Risk of Hypertension

In the next step, we evaluated the relationships between individual SES readings (education and income) and the risk of hypertension. While carrying out this stratified analysis, we controlled the influence of confounding variables and determined the strength of the association between hypertension and, separately, education level and income (Table 4).

In this logistic regression analysis, the referent group was comprised of participants with higher education whose income was higher than 400 euros. Among the participants of this group, the prevalence of hypertension was 27.2%, while among the participants with higher education whose monthly income was below 400 euros, the prevalence of hypertension was 58.3%. The multivariate logistic regression models showed that, irrespective of income, lower education tended to increase odds ratios for hypertension by about 50%. However, low income among the participants with higher education was associated with a significant increase in the risk of hypertension in the unadjusted (OR 3.76, 95% CI 1.59–8.88) and the adjusted (OR 2.81, 95% CI 1.02–7.74) models. These results revealed that both education level and income contributed to the disparities in the risk of hypertension in different SES groups.

Subsequently, using stratified analysis and the logistic regression models, we tested the hypothesis that the differences in the effect of the social environment on hypertension were mediated by exposure to urban green spaces (Table 5). In this stratified analysis, the referent group was higher educated participants residing in green areas (NDVI ≥ mean).

Compared to the referent group, higher educated participants residing in low greenness areas (NDVI < mean) had an increased risk of hypertension in the unadjusted model (OR 1.45, 95% CI 0.82–2.56), as well as in the model adjusted for sex, age, family status, and smoking status (OR 1.65, 95% CI 0.88–3.08). A similar effect was found in the group of -lower educated participants residing in high greenness areas (NDVI > mean), albeit these results lack statistical significance. However, our findings showed a significant increase in the probability of hypertension among less educated participants residing in low greenness areas (NDVI < mean) in both the unadjusted (OR 1.75, 95% CI 1.02–2.99) and the adjusted (OR 1.83, 95% CI 1.01–3.36) models. These associations were robust to a sensitivity analysis after adjustment for sex, age, family status, smoking status, and income. The results reflect the complex interaction of the SES and the natural environment that mediates the associations. These results show good potential for reducing the risk of hypertension in the less educated population by up to 83% through the improvement of the quality of residential environment by increasing urban green spaces.

## 4. Discussion

Citizen science is an important means to promote sustainable development policy by tailoring people’s needs and demands for the promotion of health and well-being [2,31]. The results of this cross-sectional study presented data for sustainable development to ensure sustainable cities and communities, and to promote well-being for all. The findings suggest that city infrastructure facilities should be designed to ensure accessibility to green spaces. The results may serve as a basis for health promotion in urban environments via urban planning and design. We found relationships of health indicators with social indicators and green space indicators (NDVI), which highlight the need for complex studies in order to present data for city planning and policy making.

This citizen science research in which the general public of Kaunas city participated, created an opportunity for the participants to familiarize themselves with research planning and to raise societal awareness about the links between environmental issues and health. Although there is growing research evidence on the effect of the SES on health, citizen science studies that focus on these issues are sparse [32]. Meanwhile, citizens’ health literacy may support the promotion and maintenance of psychological well-being and healthy behavior such as physical activity and reduction of smoking and may improve self-reported health [33,34,35,36]. To date, joint studies on the influence of the social environment (comprising personal and societal relationships, physical surroundings) and the perception of the quality of built neighborhood, urban green spaces, and hypertension have not received adequate attention. The present citizens’ study attempted to fill this gap by, for the first time, including citizens’ concerns in the study design and the formulation of the research question when investigating the joint effects of education and income on hypertension and studying the impact of the social environment depending on exposure to green spaces.

The obtained new results of our cross-sectional study suggest that the participants’ concerns and neighborhood quality rating did not depend on the education level. However, those study participants with lower education and those study participants with higher education on low incomes rated their health significantly worse in self-reported health measures.

We observed education-related inequalities in exposure to residential green spaces of the built environment: better-educated participants resided in greener areas and more often regularly visited the natural environment. However, a significantly higher risk of hypertension was found among higher educated participants with low income (OR 2.81, 95% CI 1.02–7.74). These associations were robust to a series of sensitivity analyses. Low-SES persons residing in area with low exposure to green spaces had a significantly higher risk of hypertension when sex, age, family status, smoking, and income were accounted for (OR 1.83, 95% CI 1.01–3.36). In addition, we observed that green spaces modified the association between SES and hypertension. These data provide evidence that the social environment and the quality of built neighborhood had a complex effect on disparities in the risk of hypertension. This study findings complement the existing research on social disparities, green space, and the risk of hypertension.

To our knowledge, this is one of the first citizen science environmental epidemiological studies to report the association between the SES, the participants’ concerns and perceptions of the quality of residence neighborhoods, exposure to neighborhood green spaces, and hypertension in an Eastern European country. We tested the hypothesis that differences in the effect of the social environment on blood pressure and hypertension were mediated by exposure to urban green spaces. Our hypothesis was partly confirmed, showing that subjects with lower individual SES and a poorer residential neighborhood benefited more from urban green spaces. There are some data indicating that the level of socioeconomic development in different European countries has an impact on the area-level and individual-level socioeconomic characteristics and social exclusion of at-risk populations [37]. In order to identify the perceived quality of the residential environment, we studied satisfaction of citizens with the residential district in terms of environmental quality, infrastructure, and safety. We also identified concerns and possibilities of social relations within the district - social communication and the leisure time facilities, as well as the connection of citizens to public transport. The findings of our study revealed that the residents of Kaunas highly scored the quality of their residential neighborhood, specifically noting that public transport met their needs in the district, and that they had good opportunities for walking to reach the city’s green spaces or parks. Data from the PHENOTYPE project showed that the association between different health indices and the NDVI depends on buffer size, possibly due to the visual impact of the green space seen from home [38]. In Kaunas, about 95% of the citizens see green spaces from home and have convenient access to a city park by walking from homes for 10 min or less. Such measurement of the proximity to green urban areas was suggested as a European Commission indicator of the presence and availability of green urban areas [39]. Nevertheless, the study participants’ physical activity was low. Moderate intensity physical activity was reported by 10.7% of the participants in the lower educated group and by 12.9% of those in the higher educated group, showing that health-related behavior might be a major risk factor for health problems, including hypertension. However, higher educated participants, more often than the lower educated ones, regularly visited the natural environment. Similar results were observed in the Austrian studies [40,41]. The researchers found a perceived high social-environmental quality of the residential environment to be associated with higher levels of self-rated health and leisure-time physical activity. Both built and social environments are important determinants of physical activity, as well as of major cardiovascular risk factors.

In this study, participants in both lower and higher educated groups similarly perceived the social-environmental quality of their residential district, which suggests that the citizens’ rating of the neighborhood quality did not depend on their education level. However, self-reported health characteristics of the lower educated participants were significantly poorer compared to those of the higher educated participants: the participants of the former group demonstrated a higher exposure to behavioral risk factors (lower physical activity, overweight, and smoking) and a higher mean systolic and diastolic blood pressure, a higher prevalence of chronic diseases, and a poorer rating of health. To our knowledge, there are only a few studies on the associations between the neighborhood characteristics and residents’ perceptions of their neighborhood that analyzed the effects of resources on well-being rather than adjusting for the variables in the multivariable models [13,19,42]. Moreover, the aforementioned studies identified and assessed different variables of neighborhood and SES from those used in our study. Meanwhile, we found that participants with higher education and low income, similarly to the lower educated ones, reported health problems more often. Thus, SES-related health inequalities were found, considering such health indicators as poorer self-reported health, chronic diseases, and a worse situation at work. These findings of ours presented evidence that both low income and lower education are independent SES indicators associated with a higher prevalence of health problems and hypertension.

In this study, we acknowledged older age, overweight, low education and income, smoking, and family status as risk factors for hypertension. Similar risk factors for hypertension were identified in other studies [43,44]. The participants with hypertension more often resided on streets with heavy traffic for longer than 20 years. We also think that disparities in hypertension across the districts were due to the participants’ individual-level characteristics and neighborhood quality, since the relationship between SES, green spaces, and hypertension persisted after the adjustment for sex, age, family status, smoking status, and income. These findings are supported by experimental investigations which showed that a synergistic effect of green spaces and exercises in green urban environments had important consequences for general health and blood pressure [45,46]. This suggests that both physical activity and exposure to green spaces positively contribute to health and have a beneficial impact on blood pressure. The health-promoting effects of green spaces may be due to stress mitigation, encouragement of physical activity, and facilitation of social contact [47,48]. Our results are in line with those of the recent studies reporting different levels of associations between exposure to green spaces and hypertension [16,17,49,50,51,52]. Possible inconsistencies in the reported effects of greenness on health are thought to be due to distance, socio-cultural factors, perceived safety [53], usage of different tools to assess the availability of green spaces [54], or different study designs [55]. Findings of a recent epidemiological study suggest that additional attention to the socioeconomic environment of the neighborhood of lower-SES individuals may reduce disparities in cardiovascular health [42].

Our findings show that the implications of city planning regarding urban green space for all might have a positive effect on all citizens’ health and well-being.

## 5. Strengths and Limitations of the Study

The strengths of this citizen science study, in relation to other studies, include a multidisciplinary approach, an environmental epidemiological study design and the usage of formalized questionnaires and standard protocols. These measures ensured the quality of the data and helped to gain new knowledge and present evidence when answering the research question formulated by the participants, namely how the quality of the residential environment and SES might affect the prevalence of hypertension. Using GIS, we were able to analyze the citizens’ satisfaction with quality of the residential district by evaluating environmental -perception, infrastructure, and safety, the citizens’ concerns and possibilities of social relations within the district, and citizens connection to public transport at the individual and the district levels in a large sample of subjects. Variability due to NDVI measures was averaged out by aggregating data within a district level, which increased the strength of the associations. Moreover, in the logistic regression models, we attempted to control the studied associations for the key confounders such as sex, age, education level, smoking, and income.

However, our study also has several limitations. This study used a cross-sectional design, and thus causal relationships could not be identified. We only describe the associations, and the causality of the relationships remains to be confirmed through additional sources. We also recognize that without objective ascertainment of physical activity and places visited, we cannot estimate the actual green space use and visual contact with green spaces. Furthermore, even though we controlled associations for possible confounding variables, residual confounding by personal characteristics is possible. We acknowledge possible errors in self-reported evaluations of health problems as the result of recall bias associated with the participants’ age and social disparity [56]. In addition, the study sample included invited participants and volunteers. However, in this study, we included questions for checking the consistency in the participants’ answers on self-reported hypertension and physical activity. Moreover, we compared the data of the 45–75 year-old (mean age: 57.3 years) participants of this study with that of a cohort of 5112 randomly selected male and female (mean age: 60.4 years) inhabitants of Kaunas city [57]. We found good concordance between self-reported hypertension and objective survey data. The prevalence of hypertension was 56.0% and 60.4%, respectively. Since we obtained relatively similar results with the representative sample, our results show a possibility for the generalization of the study findings to a large extent. In this study, the estimation of the participants’ physical activity was subjective, i.e., by using a questionnaire. To ensure a better characterization of green space quality and accessibility, it is suggested to use satellite data combined with data of other analytical tools used for the estimation of indicators of surface greenness. Time-activity based exposure to green space might reduce exposure misclassification [38].

## 6. Conclusions

This citizen science study involved participants from the general public to help answer relevant scientific questions and to generate new knowledge. The major challenges of the study were overcome by good planning of the cross-sectional study and by the usage of formalized questionnaires and standard protocols. Citizens’ participation in the assessment of the quality of the urban environment and health problems made an important contribution when identifying environmental health problems. Low physical activity, poor socioeconomic situation, and low greenness levels were the determinants significantly associated with hypertension risk. The findings of this study provide evidence that the quality of the social environment and built neighborhood had a joint effect on disparities in the risk of hypertension. In future studies, the integration of objective physical activity measurements using sensors would encourage volunteer participation, reduce exposure misclassification, and allow for obtaining detailed data to increase the sensitivity of the results. Involvement of motivated long-term participants in the study and assurance of regular feedback on how their contribution is used for answering the research questions relevant for the community would contribute to the activity of the participants involved in citizen science. The implementation of a policy for the reduction of modifiable environmental risks and elimination of health disparities might produce a beneficial effect on communities’ health and well-being.

## Figures and Tables

**Figure 1 ijerph-17-07696-f001:**
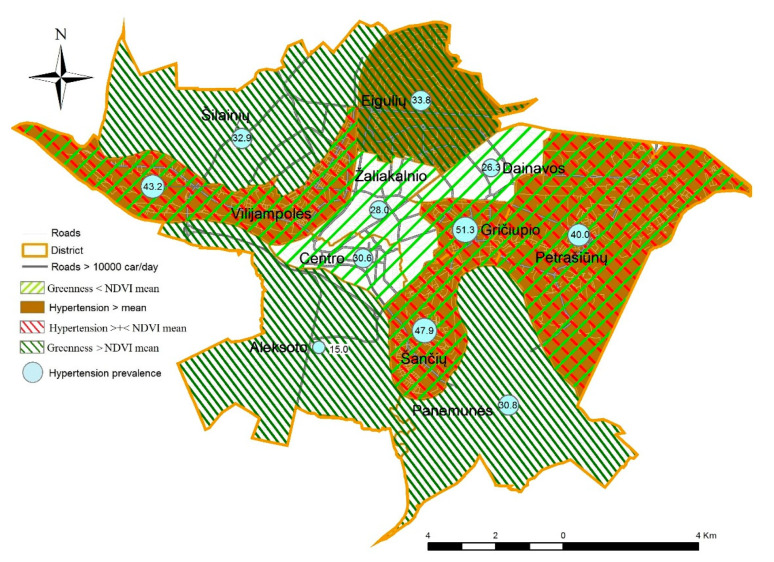
Spatial distribution of exposure to greenness (by mean) and the unadjusted prevalence of hypertension (%) at Kaunas district level.

**Table 1 ijerph-17-07696-t001:** Characteristics of the participants by education level.

Educational Status Characteristics	Lower Education, N (%)	Higher Education, N (%)	*p*
Age groups			0.354 ^‡^
18–44	195 (54.3)	164 (45.7)	
45–64	101 (57.1)	76 (42.9)	
≥65	30 (65.2)	16 (34.8)	
Sex			0.014 ^‡^
Men	152 (62.0)	93 (38.0)	
Women	174 (51.6)	163 (48.4)	
District			0.560 ^‡^
1	21 (52.5)	19 (47.5)	
2	29 (59.2)	20 (40.8)	
3	39 (52.7)	35 (47.3)	
4	50 (52.6)	45 (47.4)	
5	23 (59.0)	16 (41.0)	
6	8 (61.5)	5 (38.5)	
7	12 (52.0)	12 (48.0)	
8	33 (68.7)	15 (31.3)	
9	37 (50.7)	36 (49.3)	
10	30 (68.2)	14 (31.8)	
11	43 (52.4)	39 (47.6)	
Family status			<0.001 ^‡^
Married	145 (47.5)	160 (52.5)	
Other	181 (65.3)	96 (34.7)	
Situation at work			<0.001 ^‡^
Full-time	164 (47.5)	181 (52.5)	
Other	162 (68.4)	75 (31.6)	
Monthly net income			<0.001 ^‡^
<400 Euro	74 (75.5)	24 (24.5)	
≥400 Euro	252 (52.1)	232 (47.9)	
Traffic 10,000 cars/day			0.920 ^‡^
<10,000	255 (56.2)	199 (43.8)	
≥10,000	71 (55.5)	57 (44.5)	
Duration of living, years (mean (SE))	17.5 (0.93)	18.6 (0.94)	0.423 ^†^
NDVI			0.220 ^‡^
<mean	221 (67.8)	161 (62.9)	
≥mean	105 (32.2)	95 (37.1)	

^†^*p* value of Student’s t test; ^‡^
*p* value of the chi-squared test; SE—standard error.

**Table 2 ijerph-17-07696-t002:** Self-reported health characteristics in groups of participants by education level.

Variables	Lower Education, N (%) or Mean (SE)	Higher Education, N (%) or Mean (SE)	*p*
Body mass index (BMI)	25.4 (0.3)	24.6 (0.3)	0.051 ^†^
<30	272 (83.4)	230 (89.8)	0.029 ^‡^
BMI ≥ 30 (obesity)	54 (16.6)	26 (10.2)	
Systolic blood pressure	128 (1.0)	125 (1.0)	0.024 ^†^
Diastolic blood pressure	86 (0.8)	82 (0.7)	0.002 ^†^
Chronic disease			0.012 ^‡^
No	221 (67.8)	198 (77.3)	
Yes	105 (32.2)	58 (22.7)	
Hypertension			0.157 ^‡^
No	209 (64.1)	179 (69.9)	
Yes	117 (35.9)	77 (30.1)	
Health status			0.004 ^‡^
Good	267 (81.9)	231 (90.2)	
Poor	59 (18.1)	25 (9.8)	
Current smoking			0.006 ^‡^
No	211 (64.7)	193 (75.4)	
Yes	115 (35.3)	63 (24.6)	
Smoking duration	4.8 (0.5)	3.0 (0.4)	0.014 ^†^
Time outdoors			0.438 ^‡^
<150 min/week	291 (89.3)	223 (87.1)	
≥150 min/week	35 (10.7)	33 (12.9)	
Time in park (min/week)	21.5 (1.8)	25.7 (2.1)	0.139 ^†^

^†^*p* value of Student’s t test; ^‡^
*p* value of the chi-squared test; SE—standard error.

**Table 3 ijerph-17-07696-t003:** Mean ratings of the perceptions of neighborhood quality and social well-being by the education level.

Questions	Lower Education, Mean (SE)	Higher Education, Mean (SE)	*p*
Does the public transport in the district meet your needs?	5.42 (0.111)	5.19 (0.128)	0.163
Are you satisfied with pathways and cycling routes?	4.91 (0.122)	4.97 (0.128)	0.743
Are there opportunities for walking to reach the city’s green spaces or parks?	5.06 (0.122)	5.15 (0.132)	0.643
Do you regularly visit the natural environment?	4.00 (0.127)	4.41 (0.137)	0.031
Is there a place in your residential area adapted for exercise and relaxation?	4.49 (0.129)	4.41 (0.145)	0.662
Does air pollution in your place of residence cause problems?	4.08 (0.121)	3.79 (0.141)	0.115
Does the noise in your place of residence hinder your sleep and/or work at home?	4.73 (0.121)	4.79 (0.141)	0.750
Are there public spaces and rooms to meet people available in your residential area?	4.09 (0.125)	3.92 (0.140)	0.355
Do you feel safe in your area?	5.03 (0.115)	5.26 (0.115)	0.151
Can you take part in decision-making to improve the environment in which you live?	3.14 (0.129)	3.50 (0.143)	0.063
During the last 6 months, have you felt stress, tension, or anxiety?	4.13 (0.120)	4.32 (0.125)	0.253

All neighborhood perception scores ranged from 1 to 7: 1 = strongly disagree, and 7 = strongly agree. Higher scores indicate better neighborhood conditions.

**Table 4 ijerph-17-07696-t004:** The relationships between self-reported individual socioeconomic situation and the risk of hypertension.

Socioeconomic Situation	Hypertension No, N (%)	Hypertension Yes, N (%)	OR (95% CI)	aOR ^†^ (95% CI)
Higher education & income ≥400 €	169 (72.8)	63 (27.2)	1 (referent)	1 (referent)
Lower education & income ≥400 €	169 (67,1)	83 (32.9)	1.32 (0.89–1.95)	1.48 (0.96–2.29)
Lower education & income <400 €	40 (54.1)	34 (45.9)	2.28 (1.33–3.92)	1.50 (0.77–2.92)
Higher education & income <400 €	10 (41.7)	14 (58.3)	3.76 (1.59–8.88)	2.81 (1.02–7.74)

OR, odds ratios; aOR ^†^ adjusted for sex, age, family status, and smoking status.

**Table 5 ijerph-17-07696-t005:** Relationships between SES, exposure to greenness (NDVI), and the risk of hypertension.

NDVI & Education	Hypertension No, N (%)	Hypertension Yes, N (%)	OR (95% CI)	aOR ^†^ (95% CI)
NDVI ≥ mean & higher	71 (74.7)	24 (25.3)	1 (referent)	1 (referent)
NDVI < mean & higher	108 (67.1)	53 (32.9)	1.45 (0.82–2.56)	1.65 (0.88–3.08)
NDVI ≥ mean & lower	70 (66.7)	35 (33.3)	1.48 (0.80–2.74)	1.92 (0.95–3.85)
NDVI < mean & lower	139 (62.9)	82 (37.1)	1.75 (1.02–2.99)	1.83 (1.01–3.36)

OR, odds ratios; aOR ^†^ adjusted for: age, sex, family status, and smoking status.

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
