# Peer review of "Neighborhood Social and Built Environment and Disparities in the Risk of Hypertension: A Cross-Sectional Study"

_ijerph, 2020, doi:10.3390/ijerph17207696_

Round 1

Reviewer 1 Report

This study recruited 580 participants residing in 11 districts in Kaunas city, Lithuania, and explored the neighborhood social and build environment disparities in the risk of hypertension.

1.There are many grammar errors in this manuscript.

2.In the part of the Abstract, the author described that "Low SES persons residing in areas with low exposure to green spaces had a significantly higher risk of hypertension when sex, age, family status, smoking, and income were accounted for (OR 2.16 95% CI 1.13-4.16) ". However, the value of OR(95% CI) was inconsistent with that shown in the part of "Results" and Table 5.

  1. The introduction is too long. The author should elaborate on the key points.
  2. The study subjects were volunteers; thus, the representativeness of participants should be considered.
  3. Why do you compare the difference of selected factors between lower and higher education? The author might think that education is a confounder in assessing the risk of hypertension. However, as shown in Table 2, no significant difference in hypertension prevalence was found between lower and higher education groups.
  4. "Current smoking" was shown repeatedly in both Tables 1 and 2.

7.Table 2. Why was there an "NDVI" under the "Time in park (min/week) "?

  1. Figure 1. The author compared the prevalence of hypertension between areas. It should be noticed that this is not a population-based survey. The prevalence of hypertension can be affected by the sampled subjects. Thus, it is irrational to compare hypertension prevalence without adjusting for age and sex.
  2. In Table 1-3, the author divided the education into lower and higher levels. However, in Table 4, it was categorized as university education and lower education. Please keep consistent.
  3. In Table 5, the author described "(NDVI≥ mean). " while in the part of "Results", it was described as "(NDVI> mean). "

Reviewer 2 Report

As the authors suggest, the novelty of this study rests in its citizen science roots. The study appears to have been well-designed and executed. The science itself provides very modest, if any, new information about relationships between education or income, access to and use of green space, and hypertension. However, it is likely to be of significant interest locally. I found it very difficult to follow the introduction and this likely limited the impact of the manuscript for this reviewer.

The authors describe a project in which citizens of an urban neighborhood in a Lithuanian city were engaged in an environmental epidemiology project that aimed to put citizens’ concerns at the heart of the research agenda. The specific research study reported in the manuscript analyzed whether there exist any social differences in the effect of built neighborhood quality and exposure to urban green spaces on hypertension. As the authors state, the study had 3 specific aims:

“1) to estimate whether the citizens’ concerns and neighborhood quality rating depended on participants’ education level;

2) to assess the relationships between individual SES markers (education and income) and the risk of hypertension; and

3) to test the hypothesis that the differences in the effect of the social environment on hypertension were mediated by exposure to urban green spaces.”

Strengths: The study was well-designed. Methods were mostly well-described. Individual citizen data seemed to have been collected through a self-administered survey. These data were linked to GPS data that allowed for spatial analysis of citizens’ data. The results were well-reported. Tables and figures were helpful and appropriate. The discussion was thorough and I appreciated the thorough discussion of the study’s strengths and weaknesses.

Weaknesses

Major: The most important weakness is that the study does not provide new knowledge about relationships between education, income, hypertension, and green space. There are many published studies that have described these relationships.

Additionally, the introduction was excessively long and meandering and was very difficult to comprehend. There was a long discussion of citizen science which appeared to be largely irrelevant to the analysis reported in this manuscript and was unrelated to the title. I could not discern what the study was about.

Moderate: The methods section was similarly difficult to follow because it lacked an initial sentence or two that provided an overview. For example, there was no statement that this was an analysis of a survey of a convenience sample of residents linked to GPS data describing their neighborhoods.  Once I understood this core information, I was then able to appreciate the thorough description of the methods that followed.

The authors need to describe what new information is provided by these study results, how the study advances the field.

I recommend a major revision of the introduction. Reduce its length substantially and clarify its focus. Is this paper about citizen science or about the built environment, SES, and hypertension?

Provide a brief (1-2 sentence) overview of the approach at the beginning of the Methods section

Round 2

Reviewer 2 Report

The authors have satisfactorily addressed my two major concerns. The introduction is much improved. The unique contributions of the study are now clear to the reader.

I don't think the authors have quite addressed the concern, with which I agree, of the other reviewer, regarding their presentation of hypertension "prevalence." The information provided in the strengths and limitations paragraph regarding the authors having compared self-reported hypertension among this study's participants to that of over 5,000 randomly selected inhabitants of the same city is important and perhaps should be included in the methods. Still, this information does not support what appears to be an assumption that district-specific prevalence based on this study's participants is representative of those districts' populations.

A minor point: I don't understand the first sentence of paragraph 2 (lines 38-40). It conveys that the authors define social determinants of health as citizens' health and well-being. I suspect this is not what they mean to convey.

Author Response

Reply to the Reviewers Manuscript ID ijerph-951248

We thank the Reviewers for their suggestions how the article could be improved. We revised the manuscript taking into account comments and suggestions and made appropriate changes. We coloured the changes made in the manuscript. The reviewers’ comments have been addressed in italics.

The authors have satisfactorily addressed my two major concerns. The introduction is much improved. The unique contributions of the study are now clear to the reader.

I don't think the authors have quite addressed the concern, with which I agree, of the other reviewer, regarding their presentation of hypertension "prevalence." The information provided in the strengths and limitations paragraph regarding the authors having compared self-reported hypertension among this study's participants to that of over 5,000 randomly selected inhabitants of the same city is important and perhaps should be included in the methods. Still, this information does not support what appears to be an assumption that district-specific prevalence based on this study's participants is representative of those districts' populations.

L225-228: Moreover, we compared the data of the 45-75- year-old participants of this study with those of a randomly selected inhabitants of Kaunas city [27] and found similar results.                                                                                  We also explained district-specific hypertension prevalence of Figure1 by including:                                        L 351-355:  We explored spatial patterning in the prevalence of hypertension (in percentage) at the district level, estimated using self-reported data of study participants, and a map of Kaunas city was created (Figure 1).  The unadjusted prevalence of hypertension in various districts ranged from 15.0% to 51.3%; however, these numbers do not represent the real situation of all the population of all Kaunas districts.

L363-365: It is important to consider that disparities in the prevalence of hypertension across districts may be due to study participants’ characteristics rather than the characteristics of all districts' populations because the study did not include samples of representatives of all Kaunas districts. These data were not adjusted for possible confounding variables such as personal age, sex, education level, smoking, or income; therefore,  we were not able to explain the reasons of unequal prevalence of hypertension in Kaunas districts.

A minor point: I don't understand the first sentence of paragraph 2 (lines 38-40). It conveys that the authors define social determinants of health as citizens' health and well-being. I suspect this is not what they mean to convey.

L 38-40 now: Social determinants of health played an important role in addressing the SDG targets to promote health and well-being and to reduce inequality [11].

The manuscript once more was revised by other English Editing service.

We thank the Reviewer for the reviewing our manuscript and suggestions for improving our manuscript.

Regina Grazuleviciene